# Factors for Marketing Innovation in Portuguese Firms CIS 2014

**Patrícia Monteiro \*, Aldina Correia \*** 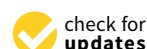 **and Vítor Braga \***

School of Management and Technology, Polytechnic of Porto, Rua do Curral, Casa do Curral, Margaride, 4610-156 Felgueiras, Portugal
**\*** Correspondence: 8150194@estg.ipp.pt (P.M.); aic@estg.ipp.pt (A.C.); vbraga@estg.ipp.pt (V.B.)

**Abstract:** Globalization, radical and frequent changes as well as the increasing importance of applying knowledge through the efficient implementation of innovation is critical under the current circumstances. Innovation has been the source of businesses competitive advantage, but it is not restricted to technological innovations, and thus marketing innovation also plays a central role. This is a significant topic in the marketing field and not yet deeply analysed in academic research. The main objective of this study is to understand what factors influence marketing innovation and to establish a business profile of firms that innovate or do not in marketing. We used multivariate statistical techniques, such as, multiple linear regression (with the Marketing Innovation Index as dependent variable) and discriminant analysis where the dependent variable is a dummy variable indicating if the firm innovates or not in marketing. The results suggest that there are several factors explaining marketing innovation, although in this study, we find that the factors contributing the most for marketing innovation are: the Organizational Innovation Index, customer and/or user suggestions, and intellectual property rights and licensing (IPRL). Most of the literature has studied these factors separately. This research studied such factors together, and it is clear that both organizational innovation and IPRL play an important role that drives firms to innovate in marketing, which differs from some literature; customer suggestions help in the process of marketing innovation, as some authors argue that customers do not always know what they want until they have it. In parallel, this study proved to be useful in understanding that the different values for the Marketing Innovation Index display no influence on the results, since they were equivalent when a dummy variable (innovated/not innovated in marketing) was used as a dependent variable. In practice, we realize that the factors are useful to clarify what Portuguese firms innovate or not in marketing, with no different results when we the four marketing innovation levels (design, distribution, advertising and price) are considered.

**Keywords:** marketing innovation; CIS 2014; multiple linear regression; discriminant analysis

## 1. Introduction

The era of globalization brought radical and frequent changes, as well as a higher recognition of the importance of knowledge through the successful implementation of innovation.

In fact, the changes are constant, and appear in different ways and at an increasing speed. These changes become a challenge for firms which need to, first, identify trends, through well-defined marketing strategies, and subsequently innovate. Innovation, according to the Organization for Economic Cooperation and Development (OECD) and Eurostat [1], requires the implementation of a new or significantly improved product (good or service), or a process, or a new marketing method, or a new organizational method in business practices, within the organization or external relations.

The role of marketing in an organization is very important since it allows increased sales by establishing a long-term relationship with customers. In fact, in addition to financial issues, marketing allows a better understanding of the customer profile leading to co-creation of value.

In order to become more competitive, firms must design new marketing approaches. Marketing innovation is considered by the literature as a non-technological innovation that lacks the same importance as technological innovations (example: product innovation) [2]. According to Mendonça et al. [3] non-technological innovation is an important factor in competitiveness and productivity growth in the economy, specifically in the service industry.

The OECD and Eurostat [1] define marketing innovation as the implementation of a new marketing concept or strategy that differs significantly from existing ones and that has not been previously used.

This study aims to gain a clearer understanding of the role of marketing innovation in Portuguese firms. First, one needs to understand which factors influence and/or impact and secondly to establish a profile of firms regarding marketing innovation.

Marketing innovation is a recent approach with a significant number of publications from 2009 [4]. Therefore, exploring what factors mostly influence Innovation in Marketing is pertinent since the literature contains limited approaches in this regard. According to Correia et al. [5], to achieve the benefits of innovation in terms of economic growth and business competitiveness, it is important to understand its determinants.

Our paper starts with a literature review, that supports the study, followed by the identification of the goals, assumptions and variables used. Subsequently, multivariate analysis of the sample taken from the CIS database (Community Innovation Survey) 2014 was performed and, finally, a connection between the literature and the results of the two statistical techniques: multiple linear regression and discriminant analysis using the SPSS (Statistical Package for the Social Sciences) are assessed.

The results suggest that there are several factors explaining marketing innovation, although, in this study, we find that the factors with higher contribution to marketing innovation are: The Organizational Innovation Index, customer and/or user suggestions and intellectual property rights and licensing (IPRL). In fact, IPRL increase the capacity of marketing innovation in the sense that firms feel more confident in sharing knowledge since they are protected [6]. In turn, the positive contribution of organizational innovation can be explained by the fact that firms increasingly apply improvements in organizational management through innovative marketing measures [7]. Finally, the contribution of customer suggestions and/or users may be related to the fact that they are the consumers of the innovations implemented through products and/or services, so they perceive of what they want to buy [8].

In parallel, this study proved to be useful for understanding that the different values for the Marketing Innovation Index display no influence on our results, since they were equivalent when using a dummy variable (innovated/not innovated in marketing) as dependent variable. In practice, we realize that such factors are useful to classify Portuguese firms that conduct marketing innovation or not, with no different results when one takes the 4 marketing innovation levels (design, distribution, advertising and price) into account.

## 2. Literature Review and Research Hypothesis

The main objective of this study is to identify the main factors that influence marketing innovation. Therefore, a survey of scientific production was conducted. Firstly, a literature review was carried out aiming to deepen the knowledge about the subject, promoting ideas for research, identifying gaps in the literature and later reviewing it, considering the methodological purpose of this study.

### 2.1. Marketing, Innovation and Marketing Innovation Concepts

Marketing is one of the most important business areas, in addition to promoting the brand of the firm, accelerating sales and business, it involves customers in the dynamics of the firm allowing a better understanding of the value proposition in a creative way. Modern consumers value the experience the

brand can provide through marketing dynamics, in contrast to the price of the product and/or service. As a result, the objective of firms is to establish a lasting relationship, giving importance to the client's opinion and involving them in the business [9].

There are numerous definitions of marketing but one of the most relevant is from the American Marketing Association [10] that defines marketing as "the activity, set of institutions, and processes for creating, communicating, delivering and exchanging offerings that have value for customers, clients, partners, and society in general". In turn, Kotler and Armstrong [11] argue that marketing is a social and management process by which individuals and organizations obtain what they need by creating and exchanging value with each other. In a restricted business context, marketing involves building profitable and valuable trading relationships with customers. Thus, the authors conceptualize marketing as the process by which firms create value and build strong relationships with customers, aiming to return this value to them [11].

Both definitions, regardless of the temporal emergence, point **customers** as focus of the firm and, consequently, marketing practices.

Dantas and Moreira [12] point out that is through innovating that one can design irreverent advertising that captivates **customers**, it allows low-price traps by competitors, namely innovation should be part of the DNA of competitive organizations. They also argue that not to innovate does not mean dying but it means being vulnerable to the most direct competitors, showing the importance of innovation to organizations.

**So, what does innovation mean?** In a Yesple way, according to the same authors, *"Innovating is creating new things, doing things differently."* The concept of innovation has been approached by several authors and it depends on its application. Table 1 points out some of existing perspectives:

**Table 1.** Innovation definitions | Source: Own Elaboration.

| Definition | Author and Year |
|---|---|
| "Innovation is defined as the formation of new products or services, new processes, raw materials, new markets and new organizations." | (Schumpeter, 1934) [13] |
| "Innovation is the specific instrument of entrepreneurship. It is the act that endows resources with a new capacity to create wealth. Innovation, indeed, creates a resource." | (Drucker, 1985) [14] |
| "Innovation is the embodiment, combination, and/or synthesis of knowledge in novel, relevant, valued new products, processes, or services." | (Leonard and Walter, 1999) [15] |
| "An innovation is the implementation of a new or significantly improved product (good or service), or process, a new marketing method, or a new organisational method in business practices, workplace organisation or external relations." | (OECD and Eurostat, 2005) [1] |
| "Innovation is the creation of something that improves the way we live our lives" | (Obama, 2007) [16] |
| "Innovation is not the result of thinking differently. It is the result of thinking deliberately (in specific ways) about existing problems and unmet needs." | (Razeghi, 2008) [17] |

In fact, these definitions are based around 3 main areas: the product (new or improved), processes and organizations (organizational innovation, management or marketing).

The OECD and Eurostat [1] present a structure (Figure 1) that shows innovation as a system and entails the different types of innovation within a firm, the connection of the firm with other organizations and the market demand.

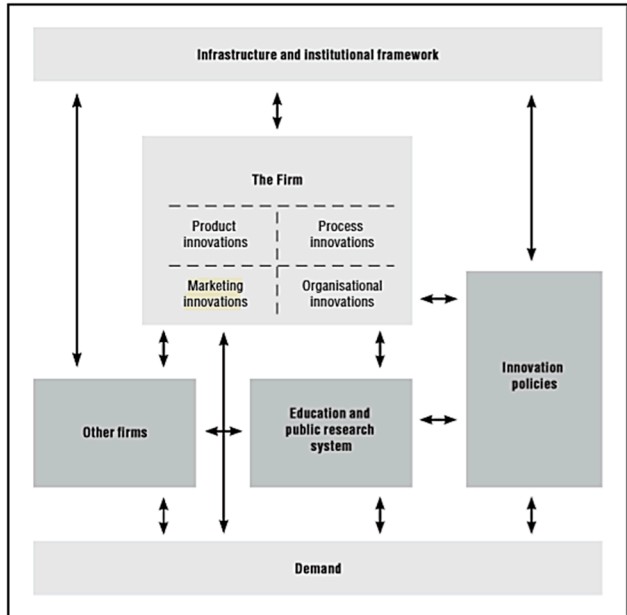

**Figure 1.** The structure of innovation | Source: [1].

The term innovation has been subject of different adjustments due to its importance in the competitive advantage of firms, thus encompassing fields beyond technological improvements, such as marketing management [18].

In fact, marketing and innovation coexist (Figure 2) and Martin [18] argues that successful modern firms are those that successfully combine innovation and marketing. For example, it is essential to firstly identify trends so that innovation can take place at a subsequent stage, considering what the market and customers need. Indeed, in recent years, new ways of collecting information about consumers through innovative marketing programs have allowed firms to reach their target audience more efficiently by using price strategies that were previously not viable [19].

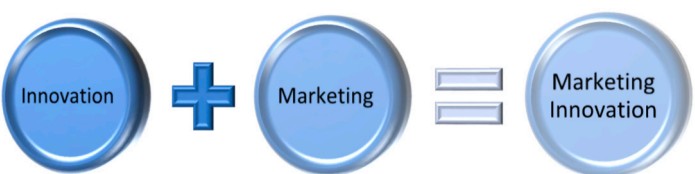

**Figure 2.** Marketing innovation | Source: Own Elaboration.

According to Hume et al. [20], Marketing Innovation develops the marketing philosophy throughout the entire innovation process that goes from the emergence of the idea (based on what the customer needs and meets their needs) to the control of the results associated to the launch of the innovation.

On the other hand, the OECD and Eurostat [1] conceptualize marketing innovation as corresponding to the implementation of a new concept or marketing strategy that differs significantly from the existing ones and that has not been previously used by firms. It requires significant changes in appearance/aesthetic or packaging, placement/distribution, promotion or on product pricing policies. It excludes seasonal changes, and regular or other routine changes in marketing methods. This definition is used throughout the present study to support our dependent variable: "Marketing Innovation Index".

*2.2. Marketing Innovation and Product Innovation (Good or Service)*

According to the OECD and Eurostat [1], product innovation corresponds to the introduction of new goods or services or significantly improved ones in the market, about their abilities or inborn abilities, ease of use, components or subsystems.

Currently, the business community strategically uses different types of innovation; one example is marketing and product innovation. The synergy between both seems to be intuitive, but there are few studies in this area. According to Gupta et al. [21], in their research on the relationship between product innovation and marketing, firms operating product innovation tend to rely on marketing as they face uncertainty about how the product will be understood by consumers. On the other hand, Junge et al. [22] concluded that firms that innovate in the product in parallel with marketing achieve a higher productivity growth. In the same line of thought, Ganzer et al. [23] tried to verify the relationship between skilled labour, turnover and number of employees with the amount of investment in product innovation, innovation process, marketing innovation and organizational innovation and concluded that firms that invest in new products or the improvement of existing products tend to innovate in marketing and, consequently, in the management of the firm. Consequently, our hypothesis is:

**Hypothesis 1.** *Product innovation contributes positively to marketing innovation.*

Instead, Rebane's [8] study shows different results, since complementarity between product innovation and marketing innovation could not be verified. However, for the services sector the results were different because service providers, when implementing innovation in services and marketing, display greater productivity. Considering these results, the following hypothesis is presented:

**Hypothesis 2.** *Innovation in services contributes positively to marketing innovation.*

*2.3. Marketing Innovation and Organizational Innovation*

The OECD and Eurostat [1] show that organizational innovation corresponds to the introduction of a new organizational method in business practices (including knowledge management), in the organization or in the firm's external relations. Higgins [24] mentions that organizational innovation is essential for firms willing to pursue strategic challenges, as they result in improvements in the management of the organization.

The relationship between Organizational Innovation and Marketing Innovation is poorly explored in the literature, but Fleacă et al. [7] studied the extent to which a marketing research process is essential in Organizational Innovation. Their article aimed to understand the importance of using well-defined processes and innovative marketing research, linking the organization's stakeholders to improve work and the overall results of the business.

Marketing research is a sub-process of marketing included in the core processes of a firm, since an effective model of market research allows an organization to more directly and economically commercialize its innovative products, according to current market trends.

The modeling marketing research workflow has drawn valuable results from the APQC (the business process classification framework that allows firms to compare their business processes with other firms [24]). Process classification frameworks developed by the worldwide leader organization in business practice, benchmarking and knowledge management [7].

In this way, a process analyst may be able to structure the necessary steps, such as research objectives, collection, methods and data analysis techniques and information to communicate their findings and implications to those responsible for organizational **decision-making** [7].

Conversely, Ganzer et al. [23] studied the relationship between: product innovation, process, marketing and organization of the knitting industry and concluded that there is a moderate positive correlation between the amount invested in product innovation with the value invested in marketing and organizational innovation. Our hypothesis is:

**Hypothesis 3.** *Innovative changes in organizational forms contribute to the innovation of marketing techniques.*

*2.4. Marketing Innovation and Suggestions of Clients and/or Users in Their Innovation Activities and in the Production of Their Innovative Goods or Services*

Clients play a key role in creating and promoting the essential conditions for an innovation project as they allow firms to better understand their needs and desires [25]. Truly, customers are often the consumers of innovations implemented through products and/or services, so they provide important insights about what the market is looking for [8].

Figure 3, proposed by Kilinc et al. [25], reinforces the literature, demonstrating the role of customers in the different stages of the innovation value chain and the impact of the primary roles customers play in the major innovation variables.

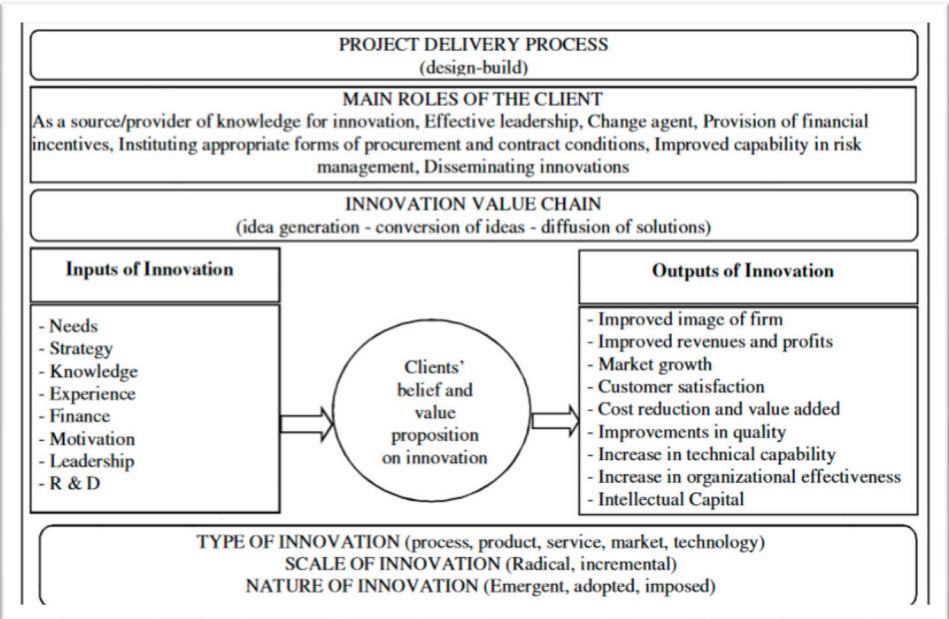

**Figure 3.** Role of clients | Prepared by: [7].

In contrast, Cabigiosu and Campagnolo [26] report that customers are a source of relevant knowledge, but cannot be used as the main or exclusive source because (i) on one hand, to develop solutions that address the specific needs of customers, there may be a limited match probability of such solutions to other market opportunities and, (ii) on the other hand, according to Tauber [27], customers often do not realize that they need certain innovative products until they are available in the market.

In fact, cooperation with customers may have a positive effect on firms; however, there are still many costs associated with cooperation with customers and negative aspects to introduce radical or revolutionary changes [8].

The literature points to the importance of customer suggestions in innovations and this article aims to understand, in addition to other factors, how customer suggestions contribute to a non-technological innovation [1], such as marketing innovation. Consequently, the following hypothesis is considered:

**Hypothesis 4.** *Customer suggestions contribute to marketing innovation.*

*2.5. Marketing Innovation and Intellectual Property Rights and Licensing*

The Oslo manual considers IPRL as requests by firms for patents, European utility models, industrial design rights and trademark registrations [1].

The connection between, for example, registration of brands and product innovation is relatively straightforward and clear, since the marketing of new products is, sometimes, associated with the creation of a new brand to communicate such innovation [3]. As far as marketing innovation is concerned, the connection between them is more complex. According to Mendonça et al. [3] amongst the four types of Innovation in the Oslo manual, only innovation in the promotion of products is not registered, all others can be registered, for example:

✳　Innovation in aesthetics, appearance and/or packaging: the famous Toblerone packaging is registered for exclusive use;
✳　Innovation in forms of distribution or sales channels of products: this type of innovation is generally not associated with a brand, except for certain firms, such as Amazon.com;
✳　Innovation in price: usually this innovation is associated with the telecommunication industry, since price is what distinguishes these types of firms.

Indeed, given the competitiveness of the market, the construction of strong brands may demand marketing innovation, in order to differ from the competition.

In their study, Olaisen and Revang [6] concluded that IPRL increases the innovation capacity, since when IPRL are in place firms feel more confident in sharing knowledge. Also, in this study it was observed that IPRL has no impact on the innovative design of the products. Therefore, the hypothesis for our study is:

**Hypothesis 5.** *Firms with intellectual property rights and licensing contribute to marketing innovation.*

*2.6. Marketing Innovation and Socioeconomic Characteristics of the Firm*

The success of innovation can be influenced by the type of organization as well as by the characteristics of its employees [21]. The success of marketing practices depends on the creation of an effective multifunctional team that works as a unit creating value for customers [28]. Consequently, the literature points out that firms involved in product innovation and marketing have qualified employees and with the adequate skills [22,29]. This indication of the literature leads to the hypothesis:

**Hypothesis 6.** *The academic degree of employees is relevant for marketing innovation.*

Employees are providers of competitive advantage for organizations, and together with turnover they define the size of businesses, i.e., whether the firm is micro, small, medium and/or large. The role of size of the firm is addressed in many studies on Innovation, since it is important to learn about their influence on marketing innovation. Sok et al. [30] state that it is essential, especially, for small and medium enterprises (SMEs) to guarantee the supply of new products, new forms/channels of distribution, to ensure customer satisfaction. The same authors further state that the Yesultaneous implementation of product innovation and marketing combined with qualified employees allow SMEs to be more competitive and achieve better results.

Another aspect leading SMEs to innovate in marketing are circumstantial austerity measures, which do not allow a more permanent support to firms. Therefore, it is imperative that SMEs maximize their internal resources and engage in marketing innovation to better understand the market [31]. In this way the following hypothesis was formulated:

**Hypothesis 7.** *The business size has an impact on marketing innovation.*

Larger firms are more likely to innovate in marketing techniques than SMEs due to the investment pressure they experience [32]. Notwithstanding the importance of the size of firms, it's also crucial to study the markets in which they operate. The market action defines the strategic path of the firm, so their decisions consider the type of market in which they choose to operate. This factor may contribute

to marketing innovation since firms are currently operating in a globalized environment, which forces them, in competitive terms, to modernize and follow the market trends [33]. Thus, we can consider the hypothesis:

**Hypothesis 8.** *Geographic markets are relevant to firms that innovate in marketing.*

Moreira [33], in his doctoral thesis on the determinants of marketing innovation, conclude that international markets display greater propensity to innovate in Marketing, however, a variable "emerge in national markets" also has a positive and significant effect on innovation marketing. Thus, we can propose the hypothesis:

**Hypothesis 9.** *Internationalization may explain marketing innovation.*

To achieve a broader explanation for the phenomenon of Marketing Innovation, we will try to understand the synergy between firms that belong to the same innovation group in marketing practices. The literature reports that the effects of synergy between firms of the same group and innovation should be treated with caution due to several factors [34].

However, through the study of Entezarkheir and Moshiri [35] it can be understood that mergers can improve incentives for innovation, promoting economies of scale, increasing the capacity to deal with uncertainty, among other things. It was also concluded that mergers are positively and significantly correlated with firm innovation. Therefore, we try to confirm that:

**Hypothesis 10.**　*Cooperation between firms of the same group is conducive to an innovative marketing environment.*

Figure 4 and Table 2 summarize the research hypothesis, pointed by literature review and considered in this work.

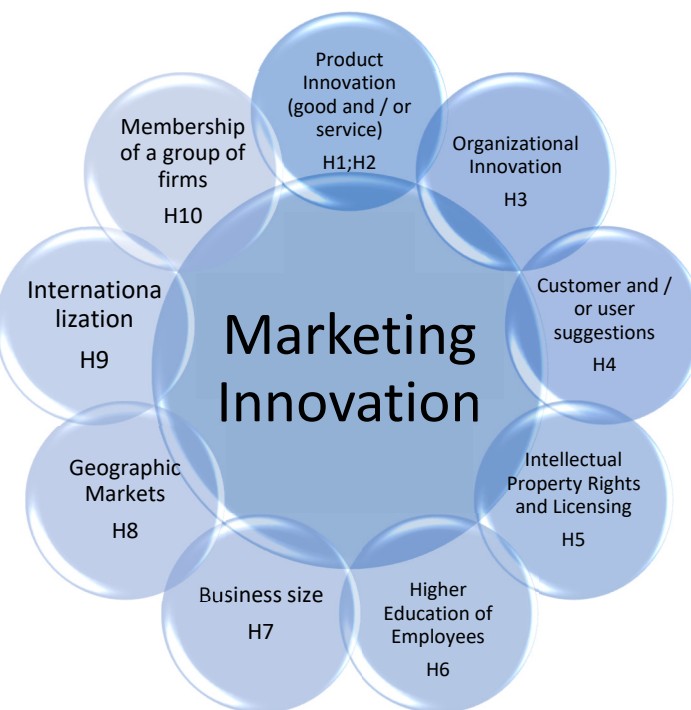

**Figure 4.** Proposed explanatory model | Source: Own Elaboration.

**Table 2.** Hypothesis synthesis and theoretical support | Source: Own Elaboration.

| Model Variables | Hypothesis | Theoretical Support |
|---|---|---|
| Product Innovation (Good and/or Service) | H1—Product innovation contributes positively to Marketing Innovation. | [8,21–23,36] |
| | H2—Innovation in services contributes positively to marketing innovation. | |
| Organizational Innovation | H3—Innovative changes in organizational forms contribute to the innovation of marketing techniques. | [7,23] |
| Customer and/or User Suggestions | H4—Customer suggestions contribute to marketing innovation. | [8,19,25–27] |
| Intellectual Property Rights and Licensing | H5—Firms that have intellectual property rights and licensing contribute to marketing innovation. | [3,6] |
| Higher Education of Employees | H6—The formation of the collaborators is relevant for the marketing innovation of a firm. | [22,29] |
| Business Size | H7—The business size has an impact on marketing innovation. | [30,31] |
| Geographic Markets | H8—Geographic markets are relevant to firms that innovate in marketing | [33,37,38] |
| Internationalization | H9—Internationalization is a factor that can help explain the phenomenon of marketing innovation. | |
| Membership of a Group of Firms (Mergers) | H10—Cooperation between firms of the same group is conducive to an innovative marketing environment. | [34,35] |

## 3. Methodology

The Community Innovation Survey (CIS) 2014 database was used for the study of Marketing Innovation. The CIS is a notation of the National Statistical System regulated by the European Union aiming to measure and characterize innovation activities in European firms. CIS 2014 covers four types of innovation: product innovation, organizational innovation, process innovation and marketing innovation, being this last innovation the focus of this study. This questionnaire is based on Eurostat guidelines and on the principles of the Oslo manual. In fact, this study, in the literature review, tried to approach the definitions contained in the manual whenever possible.

### 3.1. Population, Sample and Data Collection

The data from CIS 2014 database was the basis of our analysis. Our population was all firms located in Portugal over a period of three years, in which the sample initially consists of 8736 firms and after correction by 7083 valid firms. CIS 2014 collected data on the four types of innovation over the period 2012–2014. The database initially contained 187 variables.

### 3.2. Exploratory Analysis of Data and Study Variables

Table 3 (Frequency tables and charts in attach) presents a synthesis of the sample used in our study, which was aimed to represent and characterize the data contained in the database. Effectively, it is essential to understand our data before proceeding to multivariate statistical techniques. It can be concluded, from the analysis of Table 3, that most of the Portuguese firms in the sample did not

innovate in product, organization and marketing. Within the firms that innovate in marketing, the most frequent innovation is the innovation in the appearance/aesthetic or in the packaging of the products.

**Table 3.** Exploratory data analysis.

| | |
|---|---|
| Firms Profile 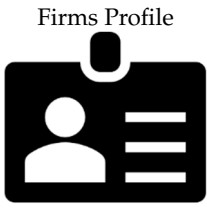 | Classification of economic activity: CAE 46, Wholesale Trade (17.5%) represent a larger share in the sample, followed by CAE 25 Manufacture of Metallic Products (8.7%) and CAE 10 Food Industries (4.5%) (Figure A2). |
| | Size: considering Decree Law 98/2015, 74.1% corresponds to small firms, 20.7% to medium-sized firms and 5.2% to large firms [39] (Table A1). |
| | Belongs to a group of firms: 71.7% of the sample, in 2014, did not belong to any group of firms (Table A2). |
| Geographic Markets 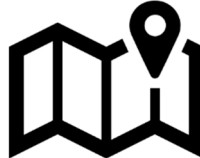 | The geographic market is another variable that is of interest for the study, with 16.5% of the sample, between 2012 and 2014, having as a geographic market to sell its goods and/or services the local/regional (MARLOC) market in Portugal, 23.4% the national market (MARNAT) in addition to the regional/local market, 24% market to the European market (MAREUR) and finally 36% to other countries not associated with the European Union (MAROTH) (Table A3). |
| Higher Education of Employees 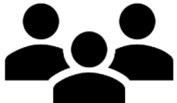 | Regarding the academic degree of the employees, 25.8% of the firms in the sample have 1 to 4% of the employees with higher education, 20.5% from 10 to 24% and 15.6% do not have any collaborators with higher education (Table A4). |
| Intellectual Property Rights and Licensing 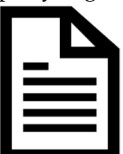 | In the scope of intellectual property and licensing, 85.9% of firms did not require any kind of intellectual property and licensing in the period from 2012 to 2014, from 14.1% requiring 11.2% acquired a patent (PROPAT), 2.2% required a European utility model (PROEUM), 0.5% registered a design right industry (PRODSG)and 0.2% registered a trademark (PROTM) (Table A5) |
| Marketing Innovation Index 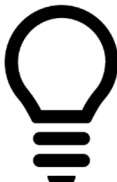 | Within the 7 083 valid firms 68.1% did not apply any type of Marketing Innovation, 13.9% applied innovations in the appearance/aesthetics or in the packaging of the products (MKTDGP), 9% in techniques or means of communication for the promotion of goods or services (MKTPDP), 5.1% in the distribution/product placement methods (goods and/or services) or new sales channels (MKTPDL) and 4% in product pricing policies (MKTPRI) (Table A6). Regarding the measures of central trend, the median is 0 meaning that 50% of the firms do not innovate in marketing. The mode is also 0, i.e., the most frequent value, explaining the 68.1% of firms that do not innovate in marketing. The standard deviation is 1.09. The Skewness/Std. Error of Skewness is 59.9, and as it is above 1.96, we conclude that the distribution of the data is asymmetric positive. The Kurtosis/Std. Error of Kurtosis is 34.86 (higher than 1.96), the data distribution is leptokurtic (Table A7). |

**Table 3.** *Cont.*

| | |
|---|---|
| Organizational Innovation Index 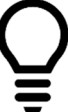 | Frequency tables show that 70.5% of the sample did not apply any new organizational method in business practices (including knowledge management), in the organization of the workplace or in the firm's external relations. Of the remaining percentage that applied, 12.7% made new business practices in the organization of procedures (ORGBUP), 9.2% applied new methods of organization of responsibilities and decision-making (ORGWKP) and finally 7.6% innovated in the methods of organization of relations external factors (ORGEXR) (Table A8). |
| Product and Service Innovation 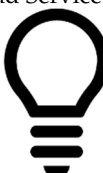 | On the other hand, most of the firms in the sample did not apply any type of innovation in both goods and services (73.5% and 81.5% respectively) (Tables A9 and A10). |
| Customer and/or User Suggestions 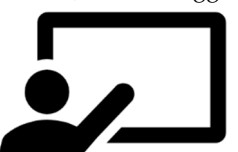 | Most of the sample did not use, during the period between 2012 and 2014, the following means of incorporating the suggestions of customers and/or users: market studies (CLUFEED), consumer groups (CLUMKT), discussion groups and interviews (CLUSUR); surveys of user needs (CLUFOR); development forums (CLUADA); and development of new goods or services by customers and/or users and that the firm has produced and introduced to the market (CLUDEV). |
| | Most of the sample <u>used</u> the following means of incorporating the suggestions of customers and/or users with the following degrees of importance: customer feedback systems (38.8%); and adaptation of existing goods or services by customers and/or users and the development, production and introduction of these goods or services on the market by the firm with a <u>medium</u> importance level with 27% (Table A11). |
| Internationalization 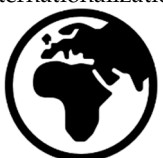 | Regarding the internationalization of the firms, 58.8% of the sample pointed out that no part of turnover results from sales to customers outside Portugal, 3.6% of firms report that 1% of turnover results from sales to customers outside Portugal and, in turn, 1.7% indicate that 100% of turnover corresponds to sales to customers outside Portugal (Figure A1). |

For this study 5 variables were created by the authors in order to investigate the validity of the research hypothesis and to provide the interpretation of the results.

Therefore, the following innovation measures were defined:

⁕ Organizational Innovation Index: this index was calculated from the dummy variables organization of procedures (ORGBUP), organization of responsibilities and decision-making (ORGWKP) and organization of relations external factors (ORGEXR) considering their sum, i.e., Inov_Org = ORGBUP + ORGWKP + ORGEXR, with values between 0 (no item selected) and 3 (all items selected) [5].

⁕ Marketing Innovation Index: this index was calculated from the dummy variables packaging of the products (MKTDGP), promotion of goods or services (MKTPDP), distribution/product placement methods (goods and/or services) or new sales channels (MKTPDL) and product pricing

policies (MKTPRI) considering their sum, i.e., Inov_Mark = MKTDGP + MKTPDP + MKTPDL + MKTPRI, with values between 0 (no item selected) and 4 (all items selected) [5].

Subsequently, the following variables were also created:

✳ Customer and/or User Suggestions, calculated considering the sum:

Sug_User = market studies (CLUFEED) + consumer groups (CLUMKT) + discussion groups and interviews (CLUSUR) + surveys of user needs (CLUFOR) + development forums (CLUADA) + development of new goods or services by customers and/or users and that the firm has produced and introduced to the market (CLUDEV)

Intellectual Property Rights and Licensing calculated considering the sum: Prop_Intellectual = acquired a patent (PROPAT) + required a European utility model (PROEUM) + registered a design right industry (PRODSG) + registered a trademark (PROTM), with values between 0 (no item selected) and 4 (all items selected).

✳ Geographic Markets: M_GEO = geographic market to sell its goods and/or services the local/regional market (MARLOC) + national market (MARNAT) + market to the European market (MAREUR) + market to other countries not associated with the European Union (MAROTH), with values between 0 (no item selected) and 4 (all items selected).

*3.3. Explanatory Variables*

Considering the data analysis and the literature review, a database was built with the variables that could allow a better understanding of Marketing Innovation. Thus, the independent variables pointed out for this multivariate study are summarized in Table 4:

**Table 4.** Explanatory Variables, Expected Signal and Theoretical Support | Source: Own Elaboration.

| Explanatory Variables | Hypothesis | Acronyms | Expected Sign | Theoretical Support |
|---|---|---|---|---|
| Product Innovation | H1 | INPDGD | + | [8,21–23,36] |
| | H2 | INPDSV | + | |
| Organizational Innovation | H3 | Inov_Org | + | [7,23] |
| Customer and/or User Suggestions | H4 | Sug_Users | + | [8,19,25–27] |
| Intellectual Property Rights and Licensing | H5 | Prop_Intellectual | + | [3,6] |
| Higher Education of Employees | H6 | EMPUD | + | [22,29] |
| Business Size | H7 | SIZE14_COD | + | [30,31] |
| Geographic Markets | H8 | M_GEO | + | [33,37,38] |
| Internationalization | H9 | SLO14 | + | |
| Membership of a Group of Firms | H10 | GP | + | [34,35] |

## 4. Factors that Influence Marketing Innovation

Multiple linear regression was used for predicting the value of a variable based on the value of two or more variables [40]. The dependent variable was "Marketing Innovation Index". The variables used to predict the value of the dependent variable are the independent variables: GP—"Belonging to a Group of Firms", Inov_Org—"Organizational Innovation Index", M_GEO—"Geographic Markets", Prop_Intellectual—"Intellectual Property Rights and Licensing", Sug_Users—"Customer and/or User Suggestions", INPDGD—"Goods Innovation", INPDSV—"Service

Innovation", EMPUD—"% Of Employees with Higher Education", SLO14—"Internationalization" and SIZE14_COD—"Business Size".

Firstly, we used the forward method in which variables are introduced one by one. The first variable to be introduced is the one with the highest correlation coefficient with the dependent variable Marketing Innovation Index. Subsequently, the variables with the highest coefficient of partial correlation are introduced sequentially [41]. Once the forward analysis was performed it was concluded that the EMPUD, SIZE14_COD, M_GEO and GP variables at a significance level of 5% are not significant for the model (Appendix A Table A12). Consequently, hypothesis H6, H7, H8 and H10 are rejected, i.e., the academic level of employees, the business size, the geographic markets and the probability of belonging to a group of firms do not contribute to explain the Index of Marketing Innovation.

After this, linear regression by the stepwise method was conducted in order to eliminate these variables from the model. By the Stepwise method of the 10 independent variables initially considered, only 6 variables were used for the estimation of the model, and the EMPUD, SIZE14_COD, M_GEO and GP variables were eliminated as expected (Appendix A Table A13).

Analyzing the summary of the multiple linear regression model (Table 5) we conclude that $Ra^2 = 0.204$ so, approximately 20.4% of the Marketing Innovation Index is explained by the independent variables.

**Table 5.** Summary | linear regression.

| Model Summary | | | | |
|---|---|---|---|---|
| **Model** | **R** | **R Square** | **Adjusted R Square** | **Std. Error of the Estimate** |
| 6 | 0.453 [f] | 0.205 | 0.204 | 1.10785 |

[f] Predictors: (Constant), Inov_Org, Sug_Users, Prop_Intellectual, INPDSV, INPDGD, SLO14.

According to the analysis of the ANOVA test (Table 6), *p*-value ≈ 0.000 so, H0 is reject, then we are faced with a highly significant model in which at least one independent variable has a considerable effect on the variation of the dependent variable of marketing innovation.

**Table 6.** Analysis of variance (ANOVA) test | linear regression.

| | ANOVA | | | | | |
|---|---|---|---|---|---|---|
| | **Model** | **Sum of Squares** | **df** | **Mean Square** | **F** | **Sig.** |
| | Regression | 1146.055 | 6 | 191.009 | 155.631 | 0.000 [g] |
| 6 | Residual | 4445.368 | 3622 | 1.227 | | |
| | Total | 5591.423 | 3628 | | | |

[g] Predictors: (Constant), Inov_Org, Sug_Users, Prop_Intellectual, INPDSV, INPDGD, SLO14.

The variables Organizational Innovation Index (with a standardized coefficient of 0.258), customer suggestions and/or users (with a standardized coefficient of 0.173) and intellectual property and licensing (with a standardized coefficient of 0.147) **are those that contribute** more to explain the Index of Marketing Innovation (Table 7).

**Table 7.** Model coefficients | linear regression.

| | **Coefficients** [a] | | | | | |
| | **Model** | **Unstandardized Coefficients** | | **Standardized Coefficients** | **t** | **Sig.** |
| | | **B** | **Std. Error** | **Beta** | | |
| | (Constant) | 0.314 | 0.035 | | 8.996 | 0.000 |
| | Inov_Org | 0.300 | 0.018 | 0.258 | 16.572 | 0.000 |
| | Sug_Users | 0.046 | 0.004 | 0.173 | 10.947 | 0.000 |
| 6 | Prop_Intellectual | 0.328 | 0.034 | 0.147 | 9.675 | 0.000 |
| | INPDSV | 0.236 | 0.043 | 0.088 | 5.539 | 0.000 |
| | INPDGD | 0.229 | 0.040 | 0.091 | 5.763 | 0.000 |
| | SLO14 | −0.301 | 0.066 | −0.070 | −4.567 | 0.000 |

[a] Dependent Variable: Inov_Mark.

> Then, the adjusted model is: Inov_Mark = 0.314 + 0.258 Inov_Org + 0.173 Sug_Users + 0.147 Prop_Intellectual + 0.088 INPDSV + 0.091 INPDGD − 0.070 SLO14

These results are, to some extent, contradictory to the literature review insofar a positive sign was expected for all independent variables (Table 4).

Contrary to expectations (Table 8—NS represent non-significant in the regression model) hypothesis H1 and H10 are rejected, then there is no statistical evidence to consider Product and Organizational Innovation as a factor to Marketing Innovation, as well as H6, H7, H8 and H10.

**Table 8.** Explanatory Variables, Obtained Signal and Theoretical Support | Source: Own Elaboration.

| Explanatory Variables | Hypothesis | Acronyms | Obtained Sign | Theoretical Support |
| --- | --- | --- | --- | --- |
| **Product Innovation** | H1 | INPDGD | NS | [8,21–23,36] |
| | H2 | INPDSV | NS | |
| **Organizational Innovation** | H3 | Inov_Org | + | [7,23] |
| **Customer and/or User Suggestions** | H4 | Sug_Users | + | [8,19,25–27] |
| **Intellectual Property Rights and Licensing** | H5 | Prop_Intellectual | + | [3,6] |
| **Higher Education of Employees** | H6 | EMPUD | NS | [22,29] |
| **Business Size** | H7 | SIZE14_COD | NS | [30,31] |
| **Geographic Markets** | H8 | M_GEO | NS | [33,37,38] |
| **Internationalization** | H9 | SLO14 | - | |
| **Membership of a Group of Firms** | H10 | GP | NS | [34,35] |

As expected, organizational innovation, customer and/or user suggestions and intellectual property rights and licensing are proved to be important for increasing marketing innovation, as pointed by literature, as well as internationalization, but the latter with opposite sign to the expected. Thus, taking into account our data, the factors promoting marketing innovation are organizational innovation, customer and/or user suggestions and intellectual property rights and licensing, and internationalization are an obstacle to innovate in marketing.

### 4.1. Testing the Assumptions of Multiple Linear Regression Analysis

In order to validate the assumptions of the Multiple Linear Regression model, a residual analysis was developed. We analyzed if the residuals follow a normal distribution and had a constant variance (using KS test and dispersion diagrams) and to understand if the residuals are independent, we used the Durbin–Watson test).

Table 9 shows the summary of the multiple linear regression model and the overall adjustment statistics. The Durbin–Watson returned a value of d = 2.002, (approximate to 2) and thus the residuals are not correlated [42]. Consequently, one could proceed with multiple linear regression.

**Table 9.** Durbin–Watson test | linear regression.

| Model Summary [g] | | | | | |
|---|---|---|---|---|---|
| **Model** | **R** | **R Square** | **Adjusted R Square** | **Std. Error of the Estimate** | **Durbin-Watson** |
| 6 | 0.453 [f] | 0.205 | 0.204 | 1.10785 | 2.002 |

[f] Predictors: (Constant), Inov_Org, Sug_Users, Prop_Intellectual, INPDSV, INPDGD, SLO14. [g] Dependent Variable: Inov_Mark.

The standard predicted and residual values show approximate maximum and minimum values but are not proportional (Table 10).

**Table 10.** Residuals statistics | linear regression.

| Residuals Statistics [a] | | | | | |
|---|---|---|---|---|---|
| | **Minimum** | **Maximum** | **Mean** | **Std. Deviation** | **N** |
| Predicted Value | 0.0130 | 3.5917 | 1.0825 | 0.57608 | 4164 |
| Residual | −2.60513 | 3.45683 | −0.01193 | 1.10424 | 4164 |
| Std. Predicted Value | −1.869 | 4.498 | 0.034 | 1.025 | 4164 |
| Std. Residual | −2.352 | 3.120 | −0.011 | 0.997 | 4164 |

[a] Dependent Variable: Inov_Mark.

Through the normal P-P plot of the regression standardized residual in Figure 5, one can conclude that some points are distant from the diagonal. This may indicate that the residuals do not follow a normal distribution.

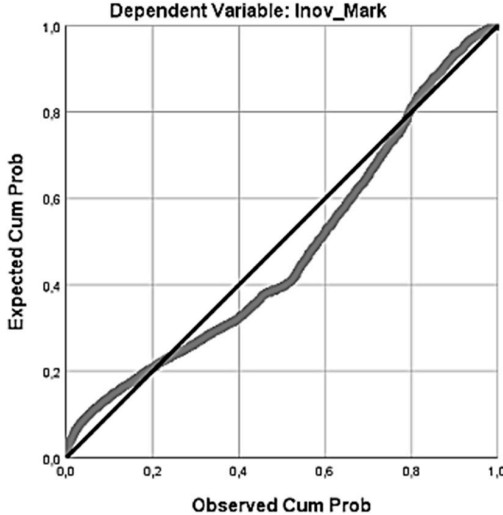

**Figure 5.** Normal P-P plot of regression standardized residual | linear regression.

In turn, the Scatterplot (Figure 6) presents horizontal lines due to the rounding errors of the values predicted by the regression model for the values of a discrete variable [28].

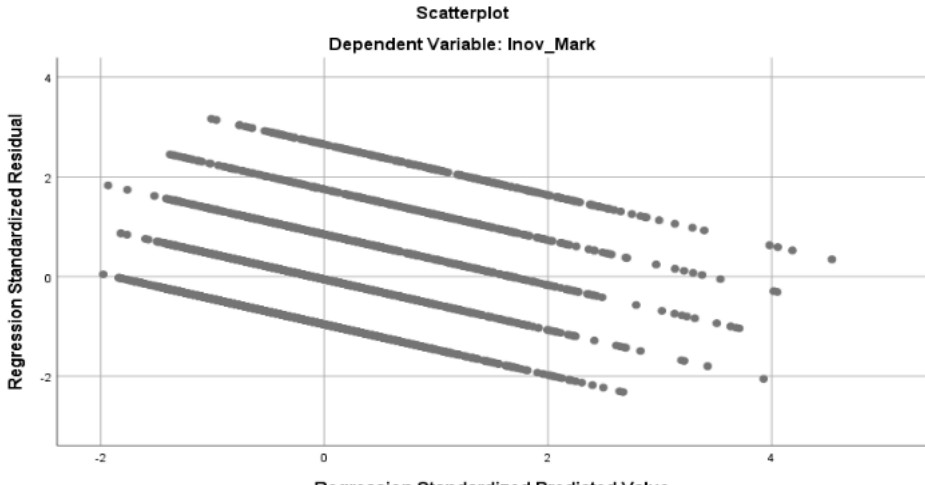

**Figure 6.** Scatterplot | linear regression.

There is an absence of correlation between independent variables (absence of multicollinearity).

Another assumption for linear regression is that none or few collinearities are present. Collinearity occurs when two independent variables are highly correlated [43].

Table 11 shows that no independent variable presents multicollinearity problems since the T is not adjacent to 0 and the Variance Inflation Factor (VIF) displays values below 5.

**Table 11.** Collinearity statistics | linear regression.

| | Coefficients [a] | | |
|---|---|---|---|
| | **Model** | **Collinearity Statistics** | |
| | | **Tolerance** | **VIF** |
| | (Constant) | | |
| | Inov_Org | 0.907 | 1.102 |
| | Sug_Users | 0.881 | 1.136 |
| 6 | Prop_Intellectual | 0.954 | 1.048 |
| | INPDSV | 0.867 | 1.153 |
| | INPDGD | 0.875 | 1.143 |
| | SLO14 | 0.928 | 1.078 |

[a] Dependent variable: Inov_Mark.

In the diagnosis of collinearity (Table 12), it follows that the values of the condition index are not close to 30 and the values themselves are distant from 0.

**Table 12.** Collinearity diagnosis | linear regression.

| | | | | Collinearity Diagnostics [a] | | | | | | |
|---|---|---|---|---|---|---|---|---|---|---|
| Model | Dimension | Eigenvalue | Condition Index | Variance Proportions | | | | | | |
| | | | | (Constant) | Inov_Org | Sug_Users | Prop_Intellectual | INPDSV | INPDGD | SLO14 |
| | 1 | 3.890 | 1.000 | 0.01 | 0.02 | 0.02 | 0.02 | 0.02 | 0.02 | 0.02 |
| | 2 | 0.875 | 2.108 | 0.00 | 0.04 | 0.00 | 0.18 | 0.20 | 0.00 | 0.35 |
| | 3 | 0.734 | 2.302 | 0.01 | 0.00 | 0.00 | 0.78 | 0.01 | 0.00 | 0.24 |
| 6 | 4 | 0.538 | 2.688 | 0.00 | 0.47 | 0.02 | 0.00 | 0.22 | 0.27 | 0.00 |
| | 5 | 0.410 | 3.082 | 0.02 | 0.01 | 0.05 | 0.02 | 0.56 | 0.37 | 0.35 |
| | 6 | 0.350 | 3.332 | 0.09 | 0.46 | 0.34 | 0.00 | 0.00 | 0.31 | 0.01 |
| | 7 | 0.202 | 4.392 | 0.87 | 0.01 | 0.57 | 0.00 | 0.00 | 0.02 | 0.03 |

[a] Dependent Variable: Inov_Mark.

Most of the proportions of variance, except for a few, show values that are distant from 50%, and may not indicate a multicollinearity problem.

Thus, generically the model meets the multiple linear regression model assumptions, and our model is significant, and there is statistical evidence in the data to consider the conclusions valid.

### 4.2. Features that Distinguish Firms that Innovate in Marketing

According to Maroco [44], discriminant analysis is "a dependent multivariate technique used to investigate, evaluate differences between groups and classify entities within groups, based on known discretionary variables." In fact, it is used to discriminate between groups, using a categorical dependent variable and independent interval scale variables [45].

As the discriminant analysis aims to discover the characteristics that distinguish the members of one group from members of a different one, the characteristics of a new individual allows predicting the group it belongs to [45]. We aimed to study which are the characteristics of firms that do not innovate in marketing and those that innovate in marketing. In particular we are interested in comparing the results with the previous analysis, where we consider marketing innovation as an index ranging between 0 (no item selected) and 3 (all items selected). For the analysis we considered Marketing Innovation as a dummy variable being 0 for non-innovative in marketing firms and 1 for innovative in marketing firms.

The non-metric dependent variable marketing innovation consists of 2 mutually exclusive categories. The independent metric variables were selected taking the literature into account. Continuously, Figure 7 presents the metric and non-metric variables under study.

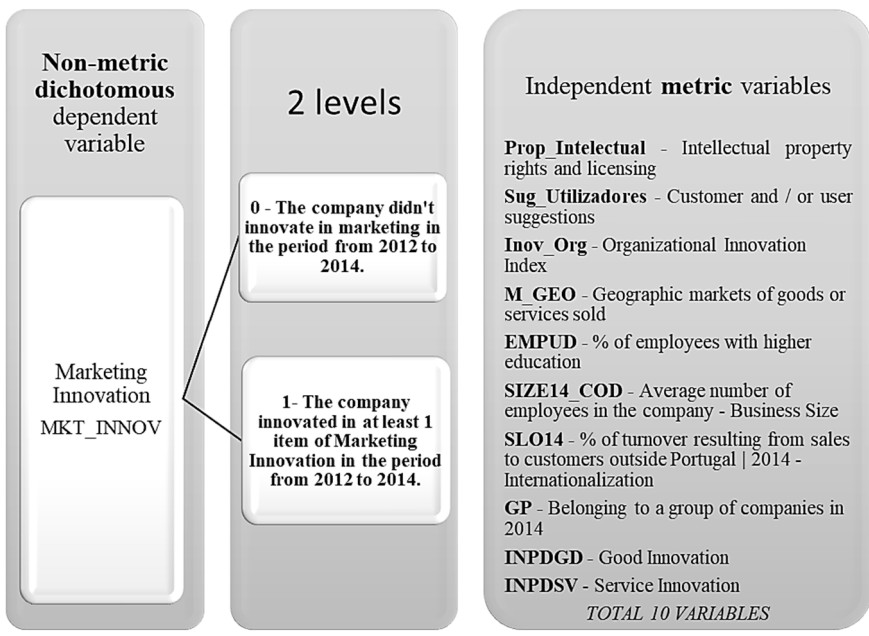

**Figure 7.** Variables metrics and non-metrics | discriminant analysis | Source: Own Elaboration.

The discriminant analysis requires the verification of the following assumptions:

1. Multivariate normality;
2. Multivariate homoscedasticity;
3. Absence of multicollinearity.

Considering the assumptions, the following tests were performed in order to understand if the discriminant analysis could be performed.

### 4.2.1. Multivariate Normality

In relation to the first assumption, a K-S test was previously developed and H0 rejected, indicating that the variables do not follow a normal distribution. In order to overcome this problem, we used the central limit theorem which indicates that the larger the size of a sample, the distribution of the mean will be closer to a normal distribution. In this case, the sample contains more than 30 cases so the distribution of the mean can be satisfactorily approximated by a normal distribution [44,46]. The remaining assumptions will then be verified in the output of the discriminant analysis.

### 4.2.2. Analysis of Variance (ANOVA): Analysis of Differences between Groups

Hypothesis to be tested:

H0: The group averages are equal
H1: The group averages are different

Looking at the test of equality of the groups means, it can be concluded that the Wilks' λ is generally approximate to 1 suggesting that the groups means are equal (Table 13).

**Table 13.** Tests of equality of group means | discriminant analysis.

| Tests of Equality of Group Means | | | | | |
|---|---|---|---|---|---|
| | Wilks' Lambda | F | df1 | df2 | Sig. |
| GP | 1.000 | 0.221 | 1 | 3627 | 0.638 |
| INPDGD | 0.985 | 54.081 | 1 | 3627 | 0.000 |
| INPDSV | 0.980 | 73.487 | 1 | 3627 | 0.000 |
| SLO14 | 0.998 | 5.547 | 1 | 3627 | 0.019 |
| SIZE14_COD | 0.999 | 1.909 | 1 | 3627 | 0.167 |
| EMPUD | 0.994 | 20.648 | 1 | 3627 | 0.000 |
| M_GEO | 0.995 | 17.135 | 1 | 3627 | 0.000 |
| Inov_Org | 0.941 | 228.403 | 1 | 3627 | 0.000 |
| Prop_Intellectual | 0.967 | 122.852 | 1 | 3627 | 0.000 |
| Sug_Users | 0.953 | 177.677 | 1 | 3627 | 0.000 |

Concerning the F-test, a small value indicates that when independent variables are considered individually, they do not differ between groups. In turn, the variable Inov_Org presents a high F suggesting being a variable that is able to differentiate the groups.

For the significance levels, most variables display a $p$-value < 0.05, thus rejecting the null hypothesis, i.e., the means in the two groups, of innovative and non-innovative firms, for all variables are equal.

In contrast with the others, the GP and SIZE14_COD present $p$-values above 5% (Table 13), indicating that these variables probably do not contribute to the model since the null hypothesis cannot be rejected.

### 4.2.3. Multivariate Homoskedasticity—Box's M test

Hypothesis to be tested:

H0: Equivalent matrices of variance–covariance for the two groups
H1: Different matrices of variance–covariance for the two groups

Analyzing the Box's M test (Table 14), it is verified that the $p$-value ≈ 0.000 < 0.05 then rejecting H0, i.e., the variance-covariance matrices are the same for the two groups. Therefore, instead of presenting homoscedasticity, required by the analysis, the data shows heteroscedasticity, becoming a limitation of the analysis.

**Table 14.** Box's M test | discriminant analysis.

| Test Results | |
| --- | --- |
| Box's M | 576.631 |
| F Approx. | 20.552 |
| df1 | 28 |
| df2 | 43125012.597 |
| Sig. | 0.000 |

Tests null hypothesis of equal population covariance matrices.

#### 4.2.4. Absence of Multicollinearity

One of the assumptions of the discriminant analysis is that there is no multicollinearity. Table 15 shows no multicollinearity, i.e., there is no high correlation between the variables, since the values are smaller than 50% presenting, in this case, levels of correlation between variables generally weak [47].

**Table 15.** Pooled within-groups matrices | discriminant analysis.

| Pooled Within-Groups Matrices | | | | | | | | | | |
| --- | --- | --- | --- | --- | --- | --- | --- | --- | --- | --- |
| | | GP | INPDGD | INPDSV | SLO14 | SIZE14_COD | EMPUD | M_GEO | Inov_Org | Prop_Intellectual | Sug_Users |
| Correlation | GP | 1.000 | 0.031 | 0.070 | 0.045 | 0.372 | 0.291 | −0.011 | 0.097 | 0.013 | 0.058 |
| | INPDGD | 0.031 | 1.000 | 0.209 | 0.185 | 0.082 | 0.007 | 0.209 | 0.051 | 0.144 | 0.174 |
| | INPDSV | 0.070 | 0.209 | 1.000 | −0.127 | 0.025 | 0.168 | −0.007 | 0.183 | 0.012 | 0.184 |
| | SLO14 | 0.045 | 0.185 | −0.127 | 1.000 | 0.266 | −0.118 | 0.257 | 0.004 | 0.118 | 0.076 |
| | SIZE14_COD | 0.372 | 0.082 | 0.025 | 0.266 | 1.000 | 0.075 | 0.086 | 0.071 | 0.075 | 0.100 |
| | EMPUD | 0.291 | 0.007 | 0.168 | −0.118 | 0.075 | 1.000 | 0.062 | 0.165 | 0.126 | 0.125 |
| | M_GEO | −0.011 | 0.209 | −0.007 | 0.257 | 0.086 | 0.062 | 1.000 | 0.016 | 0.150 | 0.130 |
| | Inov_Org | 0.097 | 0.051 | 0.183 | 0.004 | 0.071 | 0.165 | 0.016 | 1.000 | 0.048 | 0.211 |
| | Prop_Intellectual | 0.013 | 0.144 | 0.012 | 0.118 | 0.075 | 0.126 | 0.150 | 0.048 | 1.000 | 0.093 |
| | Sug_Users | 0.058 | 0.174 | 0.184 | 0.076 | 0.100 | 0.125 | 0.130 | 0.211 | 0.093 | 1.000 |

#### 4.2.5. Stepwise Method

Since, previously, it was verified that the variables GP and SIZE14_COD do not show significant discriminant power, we used the Stepwise method. This method selects the variables with discriminative capacity, so that the analysis is only done with such variables. In fact, the stepwise method starts without variables and in the following steps variables are added or removed, depending on their discriminative ability [44].

In this analysis the method used for the inclusion/removal of variables was Wilk's λ. Consequently, the variables by this method are included (or removed) according to their inclusion, it greatly decreases (or not) the lambda value [44].

By the stepwise method, only 7 (out of 10) independent variables were considered for the model estimation, and the variables GP, EMPUD and M_GEO were eliminated (Table 16).

**Table 16.** Variables not considered in the analysis | discriminant analysis.

| Variables Not in the Analysis | | | | | |
| --- | --- | --- | --- | --- | --- |
| | Step | Tolerance | Min. Tolerance | F to Enter | Wilks' Lambda |
| 7 | GP | 0.852 | 0.794 | 1.660 | 0.882 |
| | EMPUD | 0.912 | 0.850 | 0.352 | 0.882 |
| | M_GEO | 0.890 | 0.831 | 2.875 | 0.881 |

Table 17 shows that as variables were introduced, the Wilks' λ decreased. Considering that a variable with little tolerance contributes little to the model, Internationalization (SLO14), shows the smallest tolerance (0.866). Prop_Intellectual and Inov_Org are two variables that present high tolerance values which indicates that they are the ones that most contribute to the model. However, all variables present high tolerance values, thus showing their relevance to the model and the absence of multicollinearity as they approach 1.

**Table 17.** Variables in the analysis | discriminant analysis.

| | **Variables in the Analysis** | | | |
| | **Step** | **Tolerance** | **F to Remove** | **Wilks' Lambda** |
|---|---|---|---|---|
| | Inov_Org | 0.931 | 125.117 | 0.913 |
| | Sug_Users | 0.904 | 68.185 | 0.899 |
| | Prop_Intellectual | 0.964 | 81.928 | 0.902 |
| 7 | INPDSV | 0.882 | 8.671 | 0.884 |
| | SIZE14_COD | 0.917 | 8.928 | 0.884 |
| | INPDGD | 0.885 | 12.498 | 0.885 |
| | SLO14 | 0.866 | 10.370 | 0.885 |

Hypothesis to be tested:

H0: The group averages are equal

H1: The group averages are different

To understand if the functions are discriminant the Wilks' $\lambda$ test (Table 18) was performed and it was concluded that one must to reject H0, since the test shows a *p*-value below 5%, i.e., the means of the groups in the function are not equal. Therefore, the functions are discriminant.

**Table 18.** Wilks' $\lambda$ | discriminant analysis.

| | **Wilks' Lambda** | | | | | | | | |
| | | | | | | **Exact F** | | | |
| **Step** | **Number of Variables** | **Lambda** | **df1** | **df2** | **df3** | **Statistic** | **df1** | **df2** | **Sig.** |
|---|---|---|---|---|---|---|---|---|---|
| 7 | 7 | 0.882 | 7 | 1 | 3627 | 69.133 | 7 | 3621.000 | 0.000 |

**To estimate the coefficients of the discriminant function, assuring the significance of the functions**

Table 19 shows that there is 1 discriminant function and the eigenvalue attributed to function 1 is 0.134 and represents 100% of the explained variance.

**Table 19.** Eigenvalues | discriminant analysis.

| | **Eigenvalues** | | | |
| **Function** | **Eigenvalue** | **% of Variance** | **Cumulative %** | **Canonical Correlation** |
|---|---|---|---|---|
| 1 | 0.134 [a] | 100.0 | 100.0 | 0.343 |

[a] First 1 canonical discriminant functions were used in the analysis.

Regarding canonical correlation, function 1 presents a canonical correlation $(0.343)^2$ corresponding to 0.117649 so, approximately 11.8% of the variance of the groups is explained by the discriminant function 1.

**Find the contribution of the variables to the function**

Table 20 allows us to understand which variables contribute to the discriminant function. This indicates that for function 1 the variables that most contribute to distinguish innovative from non-innovative firms are Inov_Org, Prop_Intellectual and Sug_Users. On a different perspective, Organizational Innovation Index, intellectual property and licensing and suggestions of clients and/or users display a positive contribution to be classified in the group of firms that innovate in marketing.

**Table 20.** Standardized canonical discriminant functional coefficients | discriminant analysis.

| Standardized Canonical Discriminant Function Coefficients | |
|---|---|
| | **Function** |
| | **1** |
| INPDGD | 0.182 |
| INPDSV | 0.152 |
| SLO14 | −0.167 |
| SIZE14_COD | −0.151 |
| Inov_Org | 0.552 |
| Prop_Intellectual | 0.441 |
| Sug_Users | 0.416 |

In turn, the structured matrix (Table 21) allows examining the contribution (ordered by the absolute value) of each variable to the discriminant function, without the effect of collinearity. In this way, organizational innovation is the factor that most positively contributes to function 1, followed by the intellectual property rights and licensing and customer and/or user suggestions. The results, without the effect of collinearity, remained almost equal to Table 20 since the collinearity test resulted negative for the discriminant analysis.

**Table 21.** Structured matrix | discriminant analysis.

| Structure Matrix | |
|---|---|
| | **Function** |
| | **1** |
| Inov_Org | 0.686 |
| Sug_Users | 0.605 |
| Prop_Intellectual | 0.503 |
| INPDSV | 0.389 |
| INPDGD | 0.334 |
| EMPUD [a] | 0.234 |
| M_GEO [a] | 0.111 |
| SLO14 | −0.107 |
| SIZE14_COD | −0.063 |
| GP [a] | 0.036 |

[a] This variable not used in the analysis.

**Classify cases**

Table 22 allows observing coefficients by Fisher function which, in turn, allow the classification of cases into groups. Thus, it follows that the classification models:

D0 (Don't Innovate in Marketing) = −3.858 + 0.839 * INPDGD + 0.439 * INPDSV − 0.203 * SLO14 + 3.488 SIZE14_COD + 0.237 * Inov_Org − 0.096 * Prop_Intellectual + 0.155 * Sug_U

D1 (Innovate in at least 1 item of Marketing Innovation) = −4.435 + 1.109 * INPDGD + 0.681 * INPDSV − 0.627 * SLO14 + 3.309 SIZE14_COD + 0.629 * Inov_Org + 0.496 * Prop_Intellectual + 0.222 * Sug_U

**Table 22.** Classification function coefficients.

| Classification Function Coefficients | | |
|---|---|---|
| | MKT_INNOV | |
| | No | Yes |
| INPDGD | 0.839 | 1.109 |
| INPDSV | 0.439 | 0.681 |
| SLO14 | −0.203 | −0.627 |
| SIZE14_COD | 3.488 | 3.309 |
| Inov_Org | 0.237 | 0.629 |
| Prop_Intellectual | −0.096 | 0.496 |
| Sug_Users | 0.155 | 0.222 |
| (Constant) | −3.858 | −4.435 |
| Fisher's linear discriminant functions | | |

**Interpretation of the results of discrimination and validation**

Considering Table 23, 63.7% of the cases were correctly classified. In cross-validation, the percentage is almost the same (63.5%) of the original classification.

**Table 23.** Classification of results | stepwise discriminant analysis.

| Classification Results [a,c] | | | | | |
|---|---|---|---|---|---|
| | | MKT_INNOV | No | Yes | Total |
| Original | Count | No | 1010 | 647 | 1657 |
| | | Yes | 669 | 1303 | 1972 |
| | % | No | 61.0 | 39.0 | 100.0 |
| | | Yes | 33.9 | 66.1 | 100.0 |
| Cross-validated [b] | Count | No | 1006 | 651 | 1657 |
| | | Yes | 672 | 1300 | 1972 |
| | % | No | 60.7 | 39.3 | 100.0 |
| | | Yes | 34.1 | 65.9 | 100.0 |

[a] 63.7% of original grouped cases correctly classified; [b] Cross validation is done only for those cases in the analysis. In cross validation, each case is classified by the functions derived from all cases other than that case; [c] 63.5% of cross-validated grouped cases correctly classified.

In Table 24 a comparison between the linear regression model (MLR) and discriminant analysis (DA) results is presented.

As in MLR, organizational innovation, customer and/or user suggestions and intellectual property rights and licensing are proved to be important for differentiating positively innovative firms from non-innovative ones, as pointed by literature. Internalization, Yesilar to the MLR analysis, proved to be an obstacle to the promotion of marketing innovation, as much as the business size. In addition, with this DA, product innovation and organizational innovation, proved to be differentiators for distinguish marketing innovative from non-innovative firms, although not significant in differentiating the level of marketing innovation in MLR.

**Table 24.** Linear regression model vs. discriminant analysis.

| Explanatory Variables | Hypothesis | Acronyms | MLR | DA |
|---|---|---|---|---|
| Product Innovation | H1 | INPDGD | NS | + |
| | H2 | INPDSV | NS | + |
| Organizational Innovation | H3 | Inov_Org | + | + |
| Customer and/or User Suggestions | H4 | Sug_Users | + | + |
| Intellectual Property Rights and Licensing | H5 | Prop_Intellectual | + | + |
| Higher Education of Employees | H6 | EMPUD | NS | NS |
| Business Size | H7 | SIZE14_COD | NS | - |
| Geographic Markets | H8 | M_GEO | NS | NS |
| Internationalization | H9 | SLO14 | - | - |
| Membership of a Group of Firms | H10 | GP | NS | NS |

## 5. Conclusions

This study explored marketing innovation in Portuguese firms between 2012 and 2014. Two multivariate statistical techniques were performed to confirm the hypothesis resulting from the literature review, namely: multiple linear regression and discriminant analysis. Both had different objectives. In the first one, it was aimed to understand which factors contributed more to explain the Marketing Innovation Index or marketing innovation level of firms and in the second one it was aimed to define a profile of the firms that do not innovate and innovate in marketing.

Regarding multiple linear regression, it was concluded that the model is significant, and it explains 20.4% of the Marketing Innovation Index. Organizational Innovation Index, customer suggestions and/or users and IPRL were the variables with the greatest contribution to explain Marketing Innovation Index. In fact, about the contribution of IPRL, they can increase the capacity of Marketing Innovation in the sense that firms feel more confident in sharing knowledge because they are protected [6]. In turn, the positive contribution of organizational innovation can be explained by the fact that firms increasingly apply improvements in organizational management through innovative marketing measures [7]. Finally, the contribution of customer suggestions and/or users may be related to the fact that they are the consumers of the innovations implemented through products and/or services, so they have a good perception of what they want to buy [8].

Discriminant analysis reinforced the results obtained through multiple linear regression and proved useful to understand that the different indices of Marketing Innovation display no influence on the results, since they were equivalent when used a dummy variable (innovated/not innovated in marketing). In order to summarise the results of the discriminant analysis, the variables show little discriminative power, however, most of the 7,083 cases (both in the original classification and in the cross validation) were correctly classified. Product Innovation and Organizational Innovation, proved to be important to distinguish innovative from non-innovative in marketing firms, but not relevant to explain the increase of the level of marketing innovation.

Geographic markets, a higher academic level of the employees and belonging to a group of firms do not contribute to explain the Marketing Innovation, thus rejecting the hypothesis initially placed: H6, H8 and H10.

Internationalization, proved to be an obstacle to promotion of marketing innovation, as much as the business size, thus H7 and H9 are verified but with a sign different from expected in the literature.

This study offers some difficulties and limitations, namely that most of the existing literature on Marketing Innovation is considerably recent and that, since 2014 (date of the CIS database) to present, behavioral changes may occur in firms regarding the importance of marketing and innovation itself.

The results show that there is still room for exploring the factors explaining marketing innovation, and this study took some steps in this direction. In fact, a future study may consider other variables, such as cooperation, marketing activities and/or public financial support [33], since the variables used in this study although relevant, are insufficient to fully explain Marketing Innovation. In parallel, it would be relevant to obtain more recent data through primary data, for example, firm surveys, in order to enrich and complement this study or to expand the research.

**Author Contributions:** Cooperative work throughout the manuscript. All authors read and approved the final manuscript.

**Acknowledgments:** We are grateful for ESTG—P. PORTO and CIISESI for the support in the preparation of this manuscript and in the participation in SYMCOMP 2019—4th International Conference on Numerical and Symbolic Computation. Developments and Applications.

**Conflicts of Interest:** The authors declare no conflict of interest.

## Appendix A

*Appendix A.1. Frequency Tables*

**Table A1.** Business size | frequency table.

| | | Frequency | Percent | Valid Percent | Cumulative Percent |
|---|---|---|---|---|---|
| | **Size14_COD** | | | | |
| Valid | 10–49 employees | 4704 | 66.4 | 74.1 | 74.1 |
| | 50–249 employees | 1311 | 18.5 | 20.7 | 94.8 |
| | >= 250 employees | 332 | 4.7 | 5.2 | 100.0 |
| | Total | 6347 | 89.6 | 100.0 | |
| Missing | System | 736 | 10.4 | | |
| | Total | 7083 | 100.0 | | |

**Table A2.** Belonging to a group of firms | frequency table.

| | | Frequency | Percent | Valid Percent | Cumulative Percent |
|---|---|---|---|---|---|
| | **GP** | | | | |
| Valid | No | 5077 | 71.7 | 71.7 | 71.7 |
| | Yes | 2006 | 28.3 | 28.3 | 100.0 |
| | Total | 7083 | 100.0 | 100.0 | |

**Table A3.** Geographic markets | frequency table.

| | | Frequency | Percent | Valid Percent | Cumulative Percent |
|---|---|---|---|---|---|
| | **M_GEO** | | | | |
| Valid | 1.00 | 1172 | 16.5 | 16.5 | 16.5 |
| | 2.00 | 1659 | 23.4 | 23.4 | 40.0 |
| | 3.00 | 1701 | 24.0 | 24.0 | 64.0 |
| | 4.00 | 2551 | 36.0 | 36.0 | 100.0 |
| | Total | 7083 | 100.0 | 100.0 | |

**Table A4.** Higher education of employees | frequency table.

| | | Frequency | Percent | Valid Percent | Cumulative Percent |
|---|---|---|---|---|---|
| | | | | **EMPUD** | |
| Valid | 0% | 1107 | 15.6 | 15.6 | 15.6 |
| | 1%–4% | 1825 | 25.8 | 25.8 | 41.4 |
| | 5%–9% | 929 | 13.1 | 13.1 | 54.5 |
| | 10%–24% | 1451 | 20.5 | 20.5 | 75.0 |
| | 25%–49% | 770 | 10.9 | 10.9 | 85.9 |
| | 50%–74% | 495 | 7.0 | 7.0 | 92.9 |
| | 75%–100% | 506 | 7.1 | 7.1 | 100.0 |
| | Total | 7083 | 100.0 | 100.0 | |

**Table A5.** Intellectual property rights and licensing | frequency table.

| | | Frequency | Percent | Valid Percent | Cumulative Percent |
|---|---|---|---|---|---|
| | | | | **Prop_Intellectual** | |
| Valid | 0.00 | 6087 | 85.9 | 85.9 | 85.9 |
| | 1.00 | 791 | 11.2 | 11.2 | 97.1 |
| | 2.00 | 155 | 2.2 | 2.2 | 99.3 |
| | 3.00 | 38 | 0.5 | 0.5 | 99.8 |
| | 4.00 | 12 | 0.2 | 0.2 | 100.0 |
| | Total | 7083 | 100.0 | 100.0 | |

**Table A6.** Marketing Innovation Index | frequency table.

| | | Frequency | Percent | Valid Percent | Cumulative Percent |
|---|---|---|---|---|---|
| | | | | **Inov_Mark** | |
| Valid | 0.00 | 4824 | 68.1 | 68.1 | 68.1 |
| | 1.00 | 981 | 13.9 | 13.9 | 82.0 |
| | 2.00 | 638 | 9.0 | 9.0 | 91.0 |
| | 3.00 | 358 | 5.1 | 5.1 | 96.0 |
| | 4.00 | 282 | 4.0 | 4.0 | 100.0 |
| | Total | 7083 | 100.0 | 100.0 | |

**Table A7.** Measures of central tendency and asymmetry and kurtosis | descriptive analysis.

| | | GP | INPDGD | INPDSV | Inov_Mark | M_GEO | Inov_Org | Prop_Intellectual | Sug_Utilizadores | SLO14 | SIZE14_COD | EMPUD |
|---|---|---|---|---|---|---|---|---|---|---|---|---|
| | | | | | | | **Statistics** | | | | | |
| N | Valid | 7083 | 7083 | 7083 | 7083 | 7083 | 7083 | 7083 | 4164 | 7083 | 6347 | 7083 |
| | Missing | 0 | 0 | 0 | 0 | 0 | 0 | 0 | 2919 | 0 | 736 | 0 |
| Mean | | 0.28 | 0.27 | 0.18 | 0.6295 | 2.7950 | 0.5382 | 0.1783 | 6.1720 | 0.1617 | 1.31 | 2.35 |
| Median | | 0.00 | 0.00 | 0.00 | 0.0000 | 3.0000 | 0.0000 | 0.0000 | 6.0000 | 0.0000 | 1.00 | 2.00 |
| Mode | | 0 | 0 | 0 | 0.00 | 4.00 | 0.00 | 0.00 | 0.00 | 0.00 | 1 | 1 |
| Std. Deviation | | 0.451 | 0.441 | 0.388 | 1.09296 | 1.10200 | 0.94186 | 0.49279 | 4.69351 | 0.29247 | 0.565 | 1.782 |
| Skewness | | 0.963 | 1.064 | 1.624 | 1.737 | −0.331 | 1.580 | 3.381 | 0.385 | 1.745 | 1.650 | 0.473 |
| Std. Error of Skewness | | 0.029 | 0.029 | 0.029 | 0.029 | 0.029 | 0.029 | 0.029 | 0.038 | 0.029 | 0.031 | 0.029 |
| Kurtosis | | −1.074 | −0.867 | 0.639 | 2.022 | −1.256 | 1.152 | 13.876 | −0.750 | 1.636 | 1.710 | −0.754 |
| Std. Error of Kurtosis | | 0.058 | 0.058 | 0.058 | 0.058 | 0.058 | 0.058 | 0.058 | 0.076 | 0.058 | 0.061 | 0.058 |
| Minimum | | 0 | 0 | 0 | 0.00 | 1.00 | 0.00 | 0.00 | 0.00 | 0.00 | 1 | 0 |
| Maximum | | 1 | 1 | 1 | 4.00 | 4.00 | 3.00 | 4.00 | 18.00 | 1.00 | 3 | 6 |
| Percentiles | 25 | 0.00 | 0.00 | 0.00 | 0.0000 | 2.0000 | 0.0000 | 0.0000 | 2.0000 | 0.0000 | 1.00 | 1.00 |
| | 50 | 0.00 | 0.00 | 0.00 | 0.0000 | 3.0000 | .0000 | 0.0000 | 6.0000 | 0.0000 | 1.00 | 2.00 |
| | 75 | 1.00 | 1.00 | 0.00 | 1.0000 | 4.0000 | 1.0000 | 0.0000 | 10.0000 | 0.1700 | 2.00 | 4.00 |

**Table A8.** Organizational Innovation Index | frequency table.

| | | Frequency | Percent | Valid Percent | Cumulative Percent |
|---|---|---|---|---|---|
| | | | **Inov_Org** | | |
| Valid | 0.00 | 4996 | 70.5 | 70.5 | 70.5 |
| | 1.00 | 898 | 12.7 | 12.7 | 83.2 |
| | 2.00 | 653 | 9.2 | 9.2 | 92.4 |
| | 3.00 | 536 | 7.6 | 7.6 | 100.0 |
| | Total | 7083 | 100.0 | 100.0 | |

**Table A9.** Service innovation | frequency table.

| | | Frequência | Percent | Valid Percent | Cumulative Percent |
|---|---|---|---|---|---|
| | | | **INPDSV** | | |
| Valid | No | 5774 | 81.5 | 81.5 | 81.5 |
| | Yes | 1309 | 18.5 | 18.5 | 100.0 |
| | Total | 7083 | 100.0 | 100.0 | |

**Table A10.** Goods innovation | frequency table.

| | | Frequency | Percent | Valid Percent | Cumulative Percent |
|---|---|---|---|---|---|
| | | | **INPDGD** | | |
| Valid | No | 5205 | 73.5 | 73.5 | 73.5 |
| | Yes | 1878 | 26.5 | 26.5 | 100.0 |
| | Total | 7083 | 100.0 | 100.0 | |

**Table A11.** Customer and/or user suggestions | frequency table.

| | | Frequency | Percent | Valid Percent | Cumulative Percent |
|---|---|---|---|---|---|
| | | | **Sug_Users** | | |
| Valid | 0.00 | 730 | 10.3 | 17.5 | 17.5 |
| | 1.00 | 101 | 1.4 | 2.4 | 20.0 |
| | 2.00 | 276 | 3.9 | 6.6 | 26.6 |
| | 3.00 | 356 | 5.0 | 8.5 | 35.1 |
| | 4.00 | 277 | 3.9 | 6.7 | 41.8 |
| | 5.00 | 218 | 3.1 | 5.2 | 47.0 |
| | 6.00 | 352 | 5.0 | 8.5 | 55.5 |
| | 7.00 | 256 | 3.6 | 6.1 | 61.6 |
| | 8.00 | 266 | 3.8 | 6.4 | 68.0 |
| | 9.00 | 256 | 3.6 | 6.1 | 74.2 |
| | 10.00 | 211 | 3.0 | 5.1 | 79.2 |
| | 11.00 | 188 | 2.7 | 4.5 | 83.7 |
| | 12.00 | 251 | 3.5 | 6.0 | 89.8 |
| | 13.00 | 142 | 2.0 | 3.4 | 93.2 |
| | 14.00 | 97 | 1.4 | 2.3 | 95.5 |
| | 15.00 | 71 | 1.0 | 1.7 | 97.2 |
| | 16.00 | 34 | 0.5 | 0.8 | 98.0 |
| | 17.00 | 34 | 0.5 | 0.8 | 98.8 |
| | 18.00 | 48 | 0.7 | 1.2 | 100.0 |
| | Total | 4164 | 58.8 | 100.0 | |
| Omisso | System | 2919 | 41.2 | | |
| Total | | 7083 | 100.0 | | |

*Appendix A.2. Graphics*

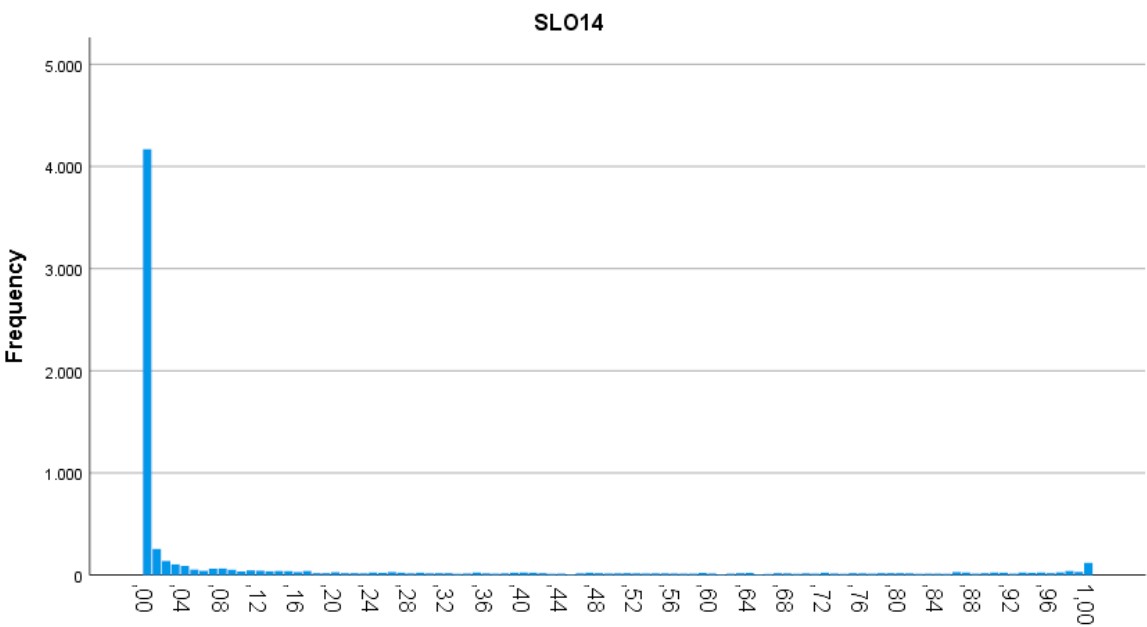

**Figure A1.** Internationalization | bar chart.

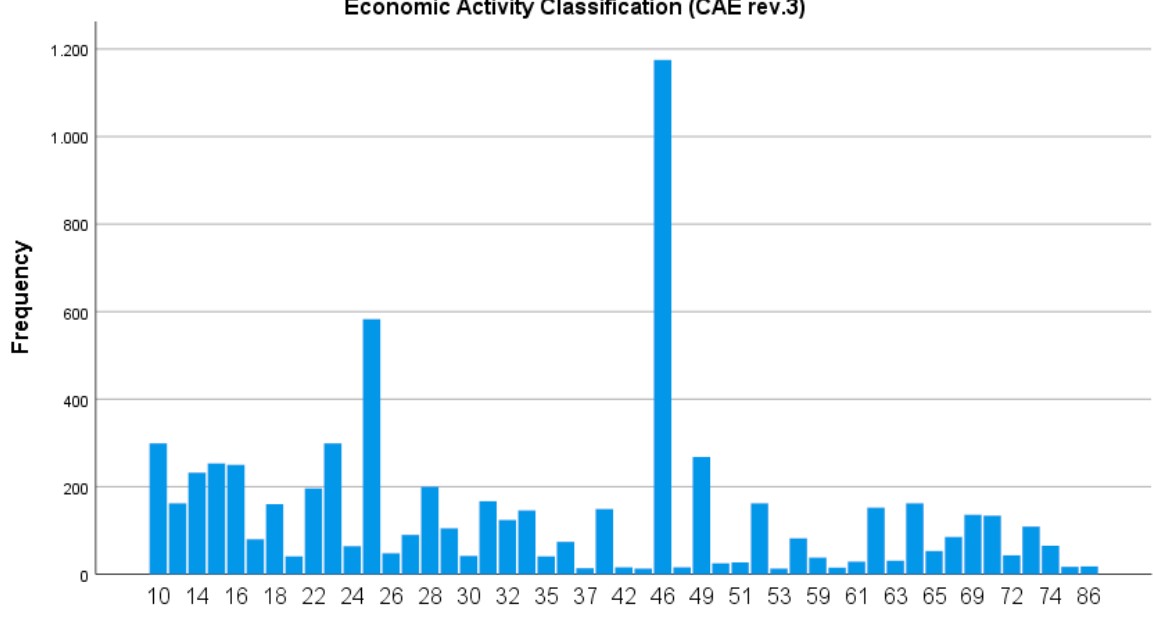

**Figure A2.** Bar chart CAE rev.3 | bar chart.

*Appendix A.3. Multiple Linear Regression | Tables*

**Table A12.** Coefficients | linear regression by forward method.

| Model | | Unstandardized Coefficients | | Standardized Coefficients | t | Sig. |
|---|---|---|---|---|---|---|
| | | B | Std. Error | Beta | | |
| | (Constant) | 0.308 | 0.072 | | 4.271 | 0.000 |
| | GP | −0.031 | 0.045 | −0.011 | −0.675 | 0.500 |
| | M_GEO | 0.018 | 0.019 | 0.015 | 0.937 | 0.349 |
| | Inov_Org | 0.302 | 0.018 | 0.259 | 16.512 | 0.000 |
| | Prop_Intellectual | 0.324 | 0.034 | 0.145 | 9.438 | 0.000 |
| 1 | Sug_Utilizadores | 0.046 | 0.004 | 0.172 | 10.827 | 0.000 |
| | SLO14 | −0.294 | 0.070 | −0.069 | −4.183 | 0.000 |
| | SIZE14_COD | −0.029 | 0.033 | −0.014 | −0.865 | 0.387 |
| | EMPUD | 0.001 | 0.012 | 0.002 | 0.128 | 0.898 |
| | INPDGD | 0.223 | 0.040 | 0.089 | 5.562 | 0.000 |
| | INPDSV | 0.240 | 0.043 | 0.089 | 5.572 | 0.000 |

[a] Dependent Variable: Inov_Mark.

**Table A13.** Variables entered/removed | Stepwise.

| Model | Variables Entered | Variables Removed | Method |
|---|---|---|---|
| 1 | Inov_Org | - | Stepwise (Criteria: Probability-of-F-to-enter <= 0.050. Probability-of-F-to-remove >= 0.100). |
| 2 | Sug_Users | - | Stepwise (Criteria: Probability-of-F-to-enter <= 0.050. Probability-of-F-to-remove >= 0.100). |
| 3 | Prop_Intellectual | - | Stepwise (Criteria: Probability-of-F-to-enter <= 0.050. Probability-of-F-to-remove >= 0.100). |
| 4 | INPDSV | - | Stepwise (Criteria: Probability-of-F-to-enter <= 0.050. Probability-of-F-to-remove >= 0.100). |
| 5 | INPDGD | - | Stepwise (Criteria: Probability-of-F-to-enter <= 0.050. Probability-of-F-to-remove >= 0.100). |
| 6 | SLO14 | - | Stepwise (Criteria: Probability-of-F-to-enter <= 0.050. Probability-of-F-to-remove >= 0.100). |

[a] Dependent Variable: Inov_Mark.

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
