# Peer review of "Factors for Marketing Innovation in Portuguese Firms CIS 2014"

_mca, doi:10.3390/mca24040099_

Round 1

Reviewer 1 Report

The objective of paper is presented clearly by raising the main question of the paper “what factors influence Marketing innovation…”. This is a significant topic in the marketing field. Nowadays there are no doubts about innovations importance for the companies but the issue about marketing innovations and their implementation isn’t analyzed deeply and academic research lacks thorough and systematic analysis in this field. It is important that this research tried to fill this gap and add valuable insights on the topic.

The suggestions to improve the paper could be as follows:

To improve the structure of the paper by depth analysis of factors which would allow not only to identify the factors but to group them as well. This would enable to systemize the theoretical and empirical results of the research and present these results more structured using figures and tables. The summary presents only the results of empirical research but what about the scientific literature analysis. What conclusions could be made? Which factors are very important for the marketing innovations and does the empirical research made in Portugal prove the research question? The significance of the research has to indicate how the insights contribute to the topic under analysis and are useful for both academics and practitioners.

Author Response

Dear reviewer, thank you for taking the time to review our article. We appreciate all suggestions. We ask the organization more time to review the English, but we send the changes made so far.

Reviewer 2 Report

General comments

The paper analyzes an interesting topic regarding to factors that influence Marketing innovation. Although the study is a large one, a number of corrections and completions are needed.

Specific comments

In the abstract there are some elements that generate confusion "this study proved to be useful to understand that the different indices of Marketing Innovation have no influence on the results". It is unclear which results are being discussed.

In the introduction there are no elements regarding the results and no comparisons with some similar studies.

Not enough studies are presented in the literature review to argue the concepts and hypotheses proposed for testing.

In fact, a number of links are analyzed in the opposite direction. In my opinion, factors such as Product Innovation, Organizational Innovation do not exert influence on Marketing Innovation, but vice versa. And the mentioned article, Fleaca (line 163), supports the relationship that marketing influences the results of the company or other processes.

Between lines 306 and 319 there are a number of models whose variables are not explained.

The results are explained mainly from a statistical point of view, without interpretations of the analyzed phenomenon and without comparisons with similar studies.

Author Response

(The authors gave the same response as above.)

Round 2

Reviewer 1 Report

Authors followed the suggestions and made the improvements.

Reviewer 2 Report

I am agree with the changes made by authors.